# Role of Internet of Things and Deep Learning Techniques in Plant Disease Detection and Classification: A Focused Review

**DOI:** 10.3390/s23187877

**Published:** 2023-09-14

**Authors:** Vijaypal Singh Dhaka, Nidhi Kundu, Geeta Rani, Ester Zumpano, Eugenio Vocaturo

**Affiliations:** 1Department of Computer and Communication Engineering, Manipal University Jaipur, Jaipur 303007, India; vijaypalsingh.dhaka@jaipur.manipal.edu; 2Sri Karan Narendra Agriculture, Jobner 303328, India; kundu.nidhi1990@gmail.com; 3Department of Informatics, Modeling Electronics and Systems (DIMES), University of Calabria, Arcavacata di Rende, 87036 Rende, Italy; e.zumpano@dimes.unical.it (E.Z.); e.vocaturo@dimes.unical.it (E.V.); 4National Research Council-Institute of Nanotechnology, Piazzale Aldo Moro, 33C, Arcavacata, 87036 Rome, Italy

**Keywords:** deep learning, convolutional neural network, precision agriculture, disease, IoT

## Abstract

The automatic detection, visualization, and classification of plant diseases through image datasets are key challenges for precision and smart farming. The technological solutions proposed so far highlight the supremacy of the Internet of Things in data collection, storage, and communication, and deep learning models in automatic feature extraction and feature selection. Therefore, the integration of these technologies is emerging as a key tool for the monitoring, data capturing, prediction, detection, visualization, and classification of plant diseases from crop images. This manuscript presents a rigorous review of the Internet of Things and deep learning models employed for plant disease monitoring and classification. The review encompasses the unique strengths and limitations of different architectures. It highlights the research gaps identified from the related works proposed in the literature. It also presents a comparison of the performance of different deep learning models on publicly available datasets. The comparison gives insights into the selection of the optimum deep learning models according to the size of the dataset, expected response time, and resources available for computation and storage. This review is important in terms of developing optimized and hybrid models for plant disease classification.

## 1. Introduction

With the increase in population, there has been a rise in the demand for agricultural products. Approximately 75% of farmers follow the traditional techniques of farming [1]. These techniques fail to meet the demands of the increasing population worldwide. The variations in climate and soil types in different regions affect crop productivity [1]. The traditional approaches do not provide any system to monitor the effects of climate and soil types. Additionally, there is no automatic mechanism for the calculation of the amount of fertilizer or pesticide required in a particular crop. This may lead to the excessive use of chemical fertilizers and pesticides. It increases the costs of agriculture, and the chemicals harm the soil as well as human health. Moreover, there is no automatic mechanism available to predict and classify plant diseases at an early stage [2]. The traditional approaches need human experts for disease detection in crops.

The above clearly shows that the high costs, need for human intervention, low yields, poor crop quality, and adverse effects of the excessive use of fertilizers and pesticides are the major challenges in traditional agriculture. There is a strong requirement to address the above-mentioned challenges. This motivated the present authors to thoroughly review the available technological solutions proposed for agriculture.

In recent years, great improvements have been observed in agriculture due to the massive enthusiasm regarding the Internet of Things (IoT) and deep learning (DL) [3,4,5,6]. IoT is useful in gathering real-time information. It helps in the judicious utilization of water, electricity, and fertilizer [7]. Further, IoT devices are efficient in monitoring visual and non-visual symptoms of disease at an early stage [8], the requirements for herbicides or pesticides, weed detection [9], and pest detection. IoT provides a fusion of imagery and parametric and genomic datasets [10]. Meanwhile, DL techniques are effective in image recognition, object detection, pattern matching, and classification [11,12]. Deep convolutional neural networks (DCNNs) are effective in automated feature extraction and feature selection. These are useful to extend the applications of DL techniques for plant disease detection [13], weed detection [14], fruit counting [15], yield prediction [16,17], and the visualization of the detected fruit, disease, or weed [15]. These techniques improve the precision and reduce the time consumed in manual feature extraction and image recognition [18,19].

The general architecture of the IoT-enabled convolutional neural network (CNN) architecture (IoTCNN) applied for the multi-class classification of plant diseases is shown in Figure 1. We considered the pearl millet plant to demonstrate its architecture [20,21]. The architecture collects the data using cameras and sensors mounted in the field. The collected data are stored on a Raspberry Pi and a cloud server. The IoTCNN helps in data analysis and decision making. Moreover, it is suitable for embedded systems and smartphones for disease detection.

The IoTCNN comprises convolution, pooling, and fully connected layers. The working mechanism of these layers is explained below.

i.
**Convolutional Layer**
This layer receives an input image in the form of a matrix. It includes a small matrix ‘kernel’ that strides over an input image to extract features from the image without destructing the spatial relationships between the pixels. The convolution operation g(x,y), as defined in Equation (Equation 1), is the dot product of two functions, h(x,y) and f(x,y). This operation is demonstrated in Figure 2.
(1)g(x,y)=h(x,y)·(x,y)A part of an input image, as shown in Figure 2 with a square box under the brown boundary shown from rows 1 to 3 and columns 1 to 3 of the input matrix, is connected to a convolutional layer to perform the convolution operation. The dot product of this part of the input image and the filter shown in Figure 2 gives a single integer of the output volume, as shown in cell (1,1) of the matrix in Figure 2. Then, the filter is moved over the next receptive field, as shown with the blue boundary from rows 1 to 3 and columns 2 to 4 in Figure 2, and performs the convolution operation again. This procedure is repeated until the filter moves over the whole image and gives a feature map, as shown in Figure 2. Different filters generate different feature maps. Therefore, the convolution layer acts as a feature detector.ii.
**Pooling Layer**
This layer receives the convoluted image as an input and applies the non-linear function. In max pooling, the maximum value of the local patch is extracted as a feature map, as shown in Figure 3. Here, 6, 7, 3, and 8 are the maximum values of features shown in the first, second, third, and fourth quadrants, respectively, in the matrix shown in Figure 3. Only these four numbers are considered in the feature map for the next step of processing. In average pooling, the average value of local patches is considered in the feature map, as shown in Figure 4. Both operations extract the relevant features from the image. Therefore, the pooling operation reduces the number of parameters and computations in a CNN model without losing vital information [22].iii.
**Fully Connected Layer**
This layer follows many convolution and pooling layers. It contains connections to all activations in the previous layer and enables the network to learn about the non-linear combinations of features for classification. It calculates the value of the gradient of the loss function and back-propagates it to the previous layers. Thus, there is a continuous update in the parameters of the model. It minimizes the value of the loss function and improves the classification accuracy.IoT devices capture climate conditions such as cloud cover, rain, sunshine, temperature, and humidity. DL techniques can work on such real-time datasets captured by IoT devices, while field monitoring and datasets collected by other devices, such as drones, cameras, etc., are used to determine the health of a crop [23]. Recording and analyzing these conditions at an early stage is significant in preventing crop disease. The smart integration of DL techniques with IoT is effective in automating disease detection, the prediction of fertilizer requirements and water requirements, crop yield prediction, etc. In this research, we provide a rigorous review of the state-of-the-art DL and IoT techniques applied in plant disease detection and classification. We also compare the performance of different DL models on the same dataset. Further, we highlight the importance of employing transfer learning and optimization techniques to achieve better performance in DL models. We also give insights for the development of an automatic tool that encompasses IoT and DL techniques to assist farmers in smart farming, as shown in Figure 5.

The rest of the paper is organized as follows. Section 2 illustrates the state-of-the-art of various deep learning models. Section 3 presents the comparative analysis of disease detection using IoT and DL. Section 4 includes the discussion. Section 5 concludes the findings and gives scope for future work. The major contributions of this manuscript are as follows:A rigorous review of the deep learning techniques used for the detection and classification of diseases in plants;The optimization of the DL models according to the response time, size of the dataset, and type of dataset;The determination of the optimum models for early disease prediction, detection, and classification;The integration of IoT and hybrid DL models for plant disease detection and classification.

## 2. State-of-the-Art Deep Learning Models

In this section, we present the deep learning models proposed in the literature for various applications.

### 2.1. AlexNet

The AlexNet model was proposed in 2012 [24] with 5-Conv and 3-FC layers. The convolution layers Conv-1 and Conv-2 are followed by normalization and a pooling layer. The last convolution layer, Conv-5, is followed by a single pooling layer, as shown in Figure 6. It introduced the use of the ReLU activation function, which improved the performance and allowed for generalization in the DL model.

The authors applied a variant of AlexNet on a maize dataset and achieved a top-five test error rate of 15.3% and accuracy of 93.8% using data augmentation and regularization methods. The model can be applied to various fields, such as plant disease detection, NLP, and medical image processing [25].

### 2.2. GoogleNet/Inception

Researchers addressed the issue of poor resource utilization and a lack of parallel processing in the AlexNet model and proposed the GoogleNet architecture in 2014 [26]. The architecture, as shown in Figure 7, has 22 layers (21-Conv and 1-FC). It has four million trainable parameters and performs batch normalization to resolve vanishing gradients. It employs the softmax activation function (σ(z)j), as defined in Equation (Equation 2). Here, *z* is the vector of the inputs given to the output layer. For example, z will contain 10 elements of 10 output units. The index of each output unit is represented by *j*, which takes values from 1 to *K*. This function is useful for multiclass classification.
(2)$$\sigma \left(z\right)j=\frac{e_{j}^{Z}}{\sum_{k=1}^{k}e^{z_{k}}}$$

GoogleNet improved the utilization of computing resources and showed a top-five test error rate of 6.67% in the ILSVRC-2014 competition, 8.63% lower than AlexNet. The model uses global average pooling instead of fully connected layers and its performance can be improved by increasing its limit of divergence [27].

### 2.3. VGGNet-16 and VGGNet-19

Although GoogleNet reduced the error rate of AlexNet, there was a scope to improve the accuracy in disease detection and classification. Researchers proposed the VGGNet-16 and VGGNet-19 models to improve the accuracy in disease detection and classification [28] in 2014. The basic architecture of the VGG model is shown in Figure 8. These models were submitted to the ILSVRC-2014 competition and achieved a top-five error rate of 7.5%. The VGG model was trained in stages to overcome vanishing gradients, and the VGG-16 model achieved 7.86% higher accuracy than AlexNet and the highest accuracy of 99.53%. VGG-19 is the most efficient classifier among AlexNet, Inception-v1, and Inception-v3, with accuracy of 99.67% [29] and a top-five error rate of 7.5%. However, it has high computation complexity due to its 130 million trainable parameters. The VGG-19 model uses the max-pooling layer to reduce the volume of the network. The authors [28] also claimed that VGG-19 is the most efficient classifier among AlexNet, Inception-v1, and Inception-v3, with accuracy of 99.67%. Moreover, its top-five error rate is 7.5%, which is 7.8% lower than that of the AlexNet model. However, it has high computation complexity due to its 130 million trainable parameters.

### 2.4. Inception-v3

For further improvements in the efficiency of CNN models, the authors of [30] used the ideas of factorization and submitted the Inception-v3 model to the ILSVRC-2015 competition. This model outperformed the benchmark classification models submitted to the ILSVRC 2012 competition. Inception-v3 contains 23 million trainable parameters. The number of parameters is smaller than that of GoogleNet, VGG-16, and VGG-19. Therefore, it requires smaller storage space. It showed a top-five error rate of 3.58% on the validation set. Its error rate is 22.42% and 11.72%, for AlexNet, GoogleNet, and VGG-19, respectively. However, while Inception-v3 achieves 2.5 times higher accuracy than GoogleNet, it is computationally expensive due to the 42 layers in the network, as shown in Figure 9.

### 2.5. ResNet

The deeper CNNs require a huge dataset for training and are difficult to train. Moreover, deep networks require a long time for training. To address these issues, researchers [31] developed residual networks in 2015. ResNet is eight times deeper than VGG-19. It contains 50, 101, or 152 layers. The basic architecture of ResNet is shown in Figure 10. ResNet-50 contains 26 million, ResNet-101 contains 60 million, and ResNet-152 contains 90 million trainable parameters. Based on the number of parameters, it is clear that ResNet-50 has low space and time complexity, whereas ResNet-152 reports the highest space and time complexity. The residual networks learn using residual functions rather than unreferenced learning. Similar to the ReLu activation function, the residual connections provide a solution for the vanishing gradient problem. The authors employed the softmax layer in ResNet to extend its applications in plant disease identification. This model reported a low error rate of 3.57%. Its error rate was 0.1% lower than that of Inception-v3. The authors submitted this CNN model to the ILSVRC-2015 and COCO 2015 challenges and competed for the first position. ResNet models require a long time for training. Thus, it becomes difficult to apply them to solve real-time problems where a quick decision is required.

### 2.6. Inception-v4 and Inception-ResNet

To further minimize the error rate and accelerate the training of the network, the authors of [32] developed a hybrid of residual and inception networks in 2016. This model has 43 million trainable parameters, as shown in Figure 11. The hybrid developed by ensembling three residuals and one Inception-v4 network gave the top-five error rate of 3.08%, which is 0.49% lower than the error rate reported for ResNet. Inception-v4 has higher space and time complexity than the ResNet-50, Inception-v3, and AlexNet models.

### 2.7. Xception

The authors of [33] replaced the inception module of the Inception model with depth-wise separable convolutions and developed the Xception architecture in 2016. It differs from the Inception model in the sequence of performing the convolution operations. Moreover, it does not employ the ReLU function for non-linearity. This model contains 23 million trainable parameters, as shown in Figure 12. The authors [33] claimed that VGG-16 [29], ResNet-152 [31], Inception-v3 [34], and Xception had top-five accuracies of 90.1%, 93.35%, 94.1%, 94.2%, and 94.5%, respectively. This shows that the Xception model outperforms the VGG-16, ResNet-152, and Inception-v3 models.

### 2.8. SqueezeNet

Developing a CNN model with low computational complexity and high accuracy has become the priority among researchers. To contribute in this direction, the authors of [35] proposed the SqueezeNet model in 2017. Its architecture is shown in Figure 13. This model uses 1 × 1 filters rather than 3 × 3 filters. The SqueezeNet model consists of a stack of fire modules and pooling layers. Each fire module contains a squeeze layer and an expanded layer. Both layers have feature maps of the same size. The squeeze layer reduces the depth of the network, whereas the expand layer increases it. This model has 1.25 million trainable parameters. The authors claimed that the SqueezeNet model is 50 times shallower than AlexNet. However, it reports top-five accuracy of 80.3%, which is equivalent to the top-five accuracy of AlexNet. Further modifications in the SqueezeNet model improved its performance. The addition of one bypass connection to the network increased its top-one accuracy by 2.9% and top-five accuracy by 2.2% [35].

### 2.9. DenseNet

In the ResNet model, the individual information is transferred to the next layer for each element. This increases the computational cost of the network. To address this challenge, researchers [36] developed the DenseNet model in 2017. Its architecture is shown in Figure 14. DenseNet architectures with 190 layers, 250 layers, and 201 layers contain 40 million, 15.3 million, and 20 million trainable parameters, respectively. In the DenseNet model, the collective information of the previous layer is transferred to the next layer. Thus, it creates direct connections among the intermediate layers and reduces the thickness of the network. This makes DenseNet more efficient than ResNet in terms of computational complexity and memory utilization. A comparative study presented by [37] showed that DenseNet performed better than ResNet and VGG models. The authors [37] claimed that the use of a customized softmax layer makes DenseNet efficient for plant disease identification. It contains a smaller number of parameters and reports high accuracy. It has shown error rates of 3.46% on C10+ and 17.18% on C100+. C10+ and C100+ are augmented forms of the CIFAR-10 and CIFAR-100 [38] datasets, respectively. The error rates were significantly lower than the error rates achieved by ResNet architectures [39].

The above discussion shows that researchers are modifying the architectures of CNN models to improve the performance, minimize the error rate, and reduce the computation time. They are also working towards extending the applicability of CNN models in different areas. They have made changes in the number of layers in the networks, activation functions, normalization strategies, and types of connections between different layers of a network. They are developing customized CNN models according to the type of dataset, size of the dataset, and nature of the problem to be solved.

Based on the discussion, it is clear that the DL models VGG [29], ResNet-01 [31], AlexNet [24], ResNet [31], and DenseNet [36] are effective in the binary as well as multiclass classification of diseases, objects, or digits, etc. However, the accuracy in multiclass classification is lower than the accuracy achieved for binary classification. Further, it is observed that the model GoogleNet is efficient in parallel computation. Therefore, it is useful in making predictions at a lower computational cost than AlexNet. It is also observed that XceptionNet is useful when higher computational costs can be endured for higher accuracy. On the other hand, the model SqueezeNet is beneficial where computational and storage resources are limited. A brief summary of the advantages and disadvantages of various DL models is given below in Table 1.

## 3. Comparative Analysis of Related Works

In this section, we give a detailed discussion of the research works conducted on applications of IoT and DL techniques in agriculture.

Diseased plants show visible lesions on the leaves or other aerial parts. These lesions vary in size, location on the plant part, shape, and color. Researchers [16,18,19,21,22] have taken advantage of the efficacy of IoT and DL techniques in pattern identification and pattern matching for the early prediction, detection, and classification of plant diseases. The sequence of steps followed for plant disease detection, visualization, and classification has been optimized.

IoT systems comprise digital and drone cameras to capture images; sensors for the monitoring of temperature, nutrients, and moisture in the soil, and humidity in the atmosphere; local processing and storage devices such as a Raspberry Pi; and cloud servers. These systems are important for the monitoring and wireless transmission of information from fields to farmers [40]. For example, the sensors mounted in the fields record any decrease in the humidity level and notify the farmer to take adequate action to prevent crop damage. Similarly, temperature sensors monitor the changes in temperature and send alerts if the temperature changes beyond a pre-set threshold. Further, the information accumulated in the sensors mounted for nutrient monitoring, weed detection, and pest detection is disseminated to farmers so that they may apply an adequate amount of fertilizer, weedicide, or pesticide. Therefore, IoT systems predominate among the traditional approaches to farming. IoT devices help by comparing the captured current parametric values of the soil and plants by the sensors with healthy plants’ values [41].

With advancements in technology, researchers have devoted great efforts to developing a complete system from data collection to plant disease prediction. The integration of IoT and DL has proven potential in developing such systems. For example, the authors of [42] integrated IoT and DL techniques for the optimal use of resources such as labor, water, chemicals, fertilizers, and pesticides. They employed ten different sensors for the monitoring of pests, soil nutrients, pH, soil air permeability, water levels, seed varieties, etc. Then, they applied the DL model to the dataset collected from the sensors and estimated the crop yield. The challenges, such as superfusion from seeding to production management and the complex backgrounds of images, reduced the efficacy of their system in real-life environments. Similarly, the authors of [43] proposed the integration of IoT and DL models for the detection of multiple diseases in crops. They employed the Multidimensional Feature Compensation–Residual Neural Network (MDFC-ResNet) and reported training accuracy of 93.96%. Next, the authors of [44] proposed the system ‘FruGar’, integrated with a web app for disease detection on coffee leaves. They employed Social IoT (SIoT) for the fusion and easy transfer of heterogeneous data comprising environmental parameters such as humidity, temperature, and pH, collected from sensors mounted at multiple locations, and images captured with a camera and smartphone. They applied the MobileNetV2 model for classification and reported accuracy of 94.58%. Then, the authors of [45,46] used sensors for the collection of parametric data and applied knowledge-based fusion techniques for feature extraction, data reduction, and classification. Their technique combined the advantages of DCNN with IoT and achieved accuracy of 97.5% in disease detection.

Based on a survey of IoT and DL-based systems, the authors of [47] claimed that the lack of sharp edges around the infected region and the impact of varying environmental conditions on image quality are major challenges in the segmentation of ROI and hence in disease prediction. Moreover, power consumption is a challenge in employing IoT techniques for crop monitoring and disease detection. The authors of [48] addressed these issues and proposed a sine cosine algorithm-based rider neural network (SCA-based RideNN). Their system employed a routing protocol to save power while transmitting images from the source to the sink. They removed the noise from images using a median filter. Their system reported accuracy of 91.18% on the PlantVillage dataset [49]. This highlights the potential to improve the accuracy of disease prediction.

Another group of researchers proposed the DL model ‘YOLOv3’ to analyze the information collected using sensors and to detect pest spots on leaves [8]. Their model also analyzed the usage of pesticides with accuracy of 90%. This is beneficial in regulating the amounts of pesticide required in a field in real time. Similarly, the authors of [50] proposed the integration of IoT and DL techniques for the monitoring of the temperature, humidity, and water supply in farms. They employed a support vector machine (SVM) and faster recurrent convolutional neural network (Faster RCNN) for the detection and classification of plant diseases. They applied K-means clustering on a dataset collected by sensors and divided the crops into six categories based on the stage of disease. The algorithm achieved accuracy of 91.5%, which is 1.15% higher than the accuracy reported in [8].

The authors in [51] explored the strengths and weaknesses of DL and IoT and assessed their impact on various aspects of modern farming. Additionally, the study investigated the potential synergies that can be achieved by integrating these technologies to maximize agricultural efficiency, sustainability, and productivity. However, the major challenge in using IoT is adverse climatic conditions and insufficient power resources for sensors. Similarly, the authors in [41] used DL and IoT for wheat rust detection.

The authors of [52] presented a review of image processing techniques applied for plant leaf disease recognition. They applied ten DL models, namely AlexNet [24], GoogleNet [26], VGG-16 [29], ResNet-101 [31], DenseNet-201 [36], Inception-v3 [30], InceptionResNet-v2 [32], Shuffle Net, SqueezeNet [35], and Mobile-Nets [53], on the PlantVillage dataset [49]. They used 54,305 leaf images from 14 species of crops. All models were trained using the same hyperparameters for 30 epochs. They employed the stochastic gradient descent with momentum (SGDm) optimizer with a pre-set value of momentum of 0.9 and a learning rate of 0.0005. The models categorized the dataset into 38 classes of diseases. A comparison of the performance of the above-stated models is demonstrated in Figure 15 using the PlantVillage data mentioned in [52]. They used a confusion matrix to evaluate the performance of DL models, as defined in Equation (Equation 3). TP denotes true positives, which represents the number of correctly classified instances of the healthy class. TN denotes true negatives, which represents the number of correctly classified instances of the diseased class. FP denotes false positives, or the number of instances of the healthy class being misclassified as the diseased class. FN denotes false negatives, which reflects the number of instances of the diseased class being misclassified as the healthy class.
(3)$$Accuracy=\frac{TP+TN}{TP+TN+FP+FN}$$

All models showed accuracy above 98.36%. Based on the performance, it is concluded that deeper models such as DenseNet-201, VGG-16, and ResNet-101 are most suitable for plant disease detection. On the other hand, ShuffleNet and SqueezeNet are efficient for embedded applications such as real-time mobile applications. However, there is room for improvement in the efficacy of these models for the accurate diagnosis of plant diseases in a short computational time.

The authors of [13] used DL techniques to develop a smartphone-assisted crop disease diagnosis system. They divided the PlantVillage dataset [49] into three sets of images: colored, grayscale, and segmented. They performed experiments on each set of images individually. They achieved accuracy of 99.35% by applying AlexNet [24] and GoogleNet [26] on the Red Green Blue (RGB) images available in the PlantVillage dataset. However, the accuracy was reduced by 31% when these models were evaluated on the real-life test dataset captured in the field. To train the models, the authors considered a dataset containing one leaf in an image. The performance of the model decreased in real life, when one image contained multiple leaves. Moreover, the authors did not focus on the visualization of the detected diseases.

The authors of [37] conducted a comparative study of CNN models proposed for plant disease detection. They trained AlexNet [24], DenseNet-169 [36], Inception-v3 [34], ResNet-34 [31], SqueezeNet-1.1 [35], and VGG-13 [29] on the PlantVillage dataset [49]. DenseNet-169 [36] and Inception-v3 [34] showed the same accuracy of 99.72%. However, Inception-v3 required a shorter training time than DenseNet-169 [36]. They addressed the problem of the visualization of the diseased parts of the plant and overcame the drawbacks identified in [13]. They used saliency maps to highlight the diseased parts of the plant. This is useful to identify the location of diseased regions without the supervision of agriculture experts. The authors claimed that visualization using saliency maps is superior to the occlusion method [37]. However, they did not provide any quantitative measurement for the visualization of the diseased parts. Moreover, saliency detection faces the issues of poor generalization [37] and poor performance.

The authors of [54] proposed a new Self-Paced Multiple-Instance Learning (SP-MIL) framework for co-saliency detection. The co-saliency helps in extracting the unique features of the images. In agriculture, it helps in the detection of diseased spots on leaves.

The authors of [55] used a combination of three-dimensional deep convolutional neural networks (3D-DCNN) and saliency maps for the early detection and clear marking of plant diseases. They collected images of plants grown after the inoculation of the causative agent of charcoal rot disease in the soil. They applied the combination of 3D-DCNN and saliency maps on 1090 images of soybean stems infected with charcoal rot disease. They extracted the spatio-temporal features from hyperspectral images with saliency mapping. These images gave the spectrum for each pixel of the diseased leaves. Therefore, the method had high computation complexity. The authors employed the Weighted Binary Cross-Entropy (WBCE) loss function in the 3D-DCNN to overcome the problem of class imbalance. The model reported the highest accuracy of 95.73%.

To achieve higher precision, researchers have developed hybrid models by combining DCNNs with detection models such as RCNNs. For example, the authors in [56] developed hybrid models using VGG-16 [29], ResNet-50, ResNet-101 [31], and Faster-RCNN for the detection of plant diseases. The experimental results showed that a combination of ResNet-101 and RCNN gave the highest average precision of 90.87% on a tomato dataset of 207 images.

From the above discussion, we conclude that the methods proposed in [8,43,50,52,54,55] for automatic disease detection are less robust. These models have reported low accuracy in disease detection when tested with variegated datasets collected from different sources. The authors of [57] worked to improve the robustness of the plant disease detection system. They developed a hybrid fully convolutional neural network (F-CNN) and segmented convolutional neural network (S-CNN) for plant disease detection and classification. They trained the model with complete images as well as segmented images. Based on the analysis of the experimental results, they claimed that the S-CNN model trained using segmented images achieved the highest accuracy of 98.6% to classify the images into ten classes. Meanwhile, the F-CNN model trained on complete images showed accuracy of 42.3%. The hybrid model was effective in classifying complete as well as segmented images.

The researchers in [4,58] applied various ML and DL models for the early prediction, detection, classification, and visualization of diseases in crops. They selected the model based on the types of parameters and the size of the dataset. A comparison of the models applied for plant disease prediction is given in Table 2.

CNN models are useful for disease detection, classification, and visualization. However, we observe that the problems of class imbalance, overfitting, underfitting, and vanishing gradients need to be addressed before implementing any model. The problem of class imbalance in the dataset can be resolved by oversampling, data augmentation [59], and assigning higher weights to the neurons for smaller samples of the dataset and lower weights to the larger samples in the dataset. The problem of overfitting can be resolved by employing regularization [24]. In regularization, the model adds an additional penalty to the value of the loss function. This minimizes the probability of obtaining extreme values of the coefficients. Hence, it controls for extreme fluctuations in the values of the loss function. The authors of [24] added dropout layers to the AlexNet model to minimize the problem of overfitting. The problem of underfitting arises when the network is shallow and it fails to capture the complex patterns available in the dataset. Researchers have increased the number of hidden layers and replaced the linear model with a non-linear model to overcome the problem of underfitting [37]. Moreover, authors have worked on the integration of CNN models and IoT devices for the early detection of disease based on parametric data sensed by IoT sensors.

### 3.1. Transfer Learning

DL models are efficient in object detection, pattern matching, and classification. However, these models need a huge amount of data for training. It is challenging to collect a labeled dataset of plant diseases at various stages of the plant life cycle. Further, the images captured at different stages of the life cycle may vary in terms of the type of background, brightness, and contrast. The differences in the quality of images may decrease the accuracy of these models. Therefore, a strong need arises to train the CNN model on a huge dataset collected from various sources. Transfer learning [60] provides a solution to this problem. In the first step, the model is trained on a large number of similar images collected from various sources. The model learns about the low-level features of the image, such as the intensity, contrast, and color of the pixels, organs, or parts plotted in the image. The weights of this trained model are saved. This saved information is used for the further training of the model. Then, the last few layers of the CNN model are trained on a dataset collected for the detection and classification of one or more diseases in a plant. At this stage, the model learns the high-level features, such as the differences in color, texture, and boundaries, distinguishing criteria between the background and the target region, etc. Based on these high-level features, the model becomes efficient in the detection and classification of diseases using different class labels. Thus, transfer learning is useful in automating the process of plant disease detection and classification. It improves the precision and reduces the training time [61].

The authors of [62] employed the concept of transfer learning for the detection of downy mildew disease in pearl millet. They applied the VGG-16 model and reported the highest accuracy of 95%. Similarly, the authors of [60] used four approaches of transfer learning with base and end-to-end CNN fine-tuning, cross-dataset fine-tuning, deep feature learning, and a convolutional neural network–recurrent neural networks (CNN-RNN) for plant classification. They evaluated the performance of all four approaches using AlexNet and VGG-16. They compared the performance of these models on four publicly available datasets: Flavia [63], the Swedish dataset [64], the UCI leaf dataset (Machine Learning Repository) [65], and the PlantVillage dataset [13]. The AlexNet and VGG-16 models reported low accuracy on the UCI leaf dataset [65] due to the small size of the dataset. These models reported high accuracy on large datasets. This proves that DL models require a large dataset size for training. Applying the cross-dataset approach yielded an improvement of 4% in accuracy. The authors of [65] claimed that the end-to-end CNN approach had the lowest accuracy when evaluated on all the datasets, namely Flavia [63], Swedish [64], the UCI leaf dataset (Machine Learning Repository) [65], and the PlantVillage dataset [13]. They observed an improvement in accuracy when the same models were trained using the concept of transfer learning. Thus, the authors [60] claimed that transfer learning is the best possible tool for prediction using DL models, even when only a small dataset is available.

### 3.2. Modified or Hybrid DL Architectures for Plant Disease Detection

According to research published by [58,66,67,68], modifications in DL architectures have been introduced for better accuracy in plant disease detection and classification. The authors of [69] proposed an improvement in the GoogleNet and Cifar-10 models. They added dropout functions, employed the ReLu activation function, and made changes in the number and positions of pooling layers in the neural network. They achieved accuracy of 98.9% on a dataset of 500 images taken from the PlantVillage dataset [49]. The accuracy was higher than that of the AlexNet and VGG models.

The authors of [70] proposed a new DL model to achieve accuracy higher than that of the SVM, AlexNet, GoogleNet, ResNet-20, and VGG-16 models for the detection of plant diseases. They rotated the images at 90°, 180°, 270°, and mirror symmetry to make the model robust to changes in the direction of the input image. They changed the contrast, sharpness, and brightness of images to decrease the light sensitivity of the model. The authors applied Gaussian noise and PCA jittering to ensure robustness against noisy datasets. In this research, they also employed Nesterov’s Accelerated Gradient (NAG) algorithm to optimize the model parameters of AlexNet [24]. The optimized model achieved accuracy of 97.62% on a dataset containing 13,689 images collected from China. The authors claimed that the optimized model yielded 10.38% higher accuracy than the AlexNet model.

Then, the authors of [71] proposed a hybrid composed of the VGG and Inception CNN architectures. Their model, ‘VGG-Inception’, reported higher accuracy than DL models, namely AlexNet, GoogleNet, ResNet, and VGG, in classifying apple plants into five disease categories. The model is useful for the clear visualization of diseases in plants. This architecture also presented inter-class detection and activation visualization. Activation visualization detects disease spots, such as grey and brown spots and rust, and separates these spots from the background images. This model is useful to differentiate diseases with a high degree of similarity in lesions. It is also efficient in the visualization of diseases identified with different class labels.

The researchers in [72] proposed DL models termed modified MobileNet and reduced MobileNet. They claimed that the reduced MobileNet model achieved the highest classification accuracy of 98.34% on a PlantVillage dataset of 82,161 images. Furthermore, the model had a smaller number of parameters than the VGG architecture. Therefore, it was successful in reducing the computational time. The architecture of the hybrid model is shown in Figure 16.

DL models are efficient in plant disease detection and classification, but the following challenges need to be addressed for automated plant disease prediction, detection, classification, and visualization.

The segmentation techniques are ineffective for background removal and in accurately extracting the ROI.The classifier reports low accuracy in classifying diseases with a high degree of similarity in symptoms.The detection of diseases using images of stems, flowers, and fruits is difficult and imprecise.The development of a a compact CNN model for mobile-embedded applications is still an unaddressed challenge.The large computational time and resource requirements of DCNN models make them less advantageous for real-life applications where a quick decision is required.

### 3.3. Optimizing the Deep Learning Models

The size and quality of the dataset, versatility in the dataset, randomly assigned weights to the neurons, the learning rate of the network, the depth of the network, the filter size selected, and the activation function are factors that greatly affect the training efficacy of a DL model. The techniques of data augmentation [59], such as rotation, shearing, and transfer learning [61,73], are useful in increasing the size of the dataset.

The training efficacy of the model can be improved by fine-tuning its hyperparameters. These are configuration variables, and their values are estimated from the training dataset. Researchers find the optimum values of these parameters by exploring the range of possible values. The optimum values of these parameters are important in solving predictive modeling problems.

It is clear from the above discussion that one architecture for the DL model and the same set of parameters is not suitable for the detection and classification of plant diseases captured in different datasets. This highlights the need to select and optimize the DL model according to the dataset available. Based on a review of DL models [31,32,36,57], we observe that the ResNet, Inception-v3, and DenseNet CNN models are useful for the classification of large datasets. A performance comparison of different DL models on tomato crop images available in the PlantVillage dataset is shown in Figure 17.

### 3.4. IoT-Enabled CNN Models Applied for Plant Disease Detection and Classification

A study of the related literature reveals that a rise in the use of DL techniques for plant disease detection and classification has been observed since 2016. The researchers in [13,37,52] applied different DL models to 54,306 images collected from the PlantVillage dataset [49] and the models were employed to recognize 14 crop species and 26 disease classes. They also used these models for the classification of images with 38 class labels. A comparative analysis of the most popularly used DL models is shown in Figure 18.

The comparative analysis of different DL models showed that in AlexNet [24], the tanh activation function was replaced with the ReLu activation function. This model was evaluated on the publicly available ImageNet dataset [74]. In 2014, further enhancements in DL models were achieved, and VGG-16, VGG-19, GoogleNet, and Inception-v3 were developed. These models are suitable for parallel computation. The efficacy in pattern matching makes these models suitable for plant disease classification. The computational time increases with an increase in the depth of the network. Therefore, to minimize the computational time, the DL models Inception-v3, ResNet-50, ResNet-101, and ResNet-152 were developed in 2015. These models support residual connections, allow batch normalization, and resolve the problem of gradient degradation.

The integration of IoT technologies in agriculture holds great promise to revolutionize farming practices, increase productivity, and address sustainability concerns. This extensive review aimed to offer insights into the current state of IoT applications in agriculture. Traditionally, the IoT architecture consists of five layers: a physical layer, network layer, middleware layer, processing layer, and application layer. By implementing an IoT layer architecture in agriculture, farmers can enhance their decision-making processes, optimize resource utilization, and ultimately increase productivity while being more environmentally sustainable. A basic outline of the IoT layer architecture for agriculture is as follows.

1.Sensors and ActuatorsVarious types of sensors are deployed throughout the agricultural fields and facilities to monitor different parameters. These may include soil moisture, temperature, humidity, light intensity, weather conditions, crop health, water level, and more. Actuators are used to control certain actions, such as turning on irrigation systems, opening and closing valves, or adjusting greenhouse environments.2.ConnectivityThis layer handles the communication between sensors, actuators, and the central system. It can use technologies such as WiFi, Bluetooth, Zigbee, LoRaWAN, or cellular networks based on the range and requirements of the specific application.3.Edge Devices/GatewaysEdge devices act as intermediaries between sensors/actuators and the cloud. They preprocess and filter the data locally, reducing the load on the central cloud infrastructure and enabling faster responses for critical tasks. Gateways facilitate communication between edge devices and the central cloud platform, ensuring data transmission and security.4.Cloud PlatformThe central cloud platform receives and stores data from multiple edge devices and sensors. It processes and analyzes the data to generate valuable insights and actionable information for farmers. Cloud-based services can include data analytics, machine learning models, and historical data storage.5.Data Analytics and Machine/Deep LearningAdvanced data analytics techniques and machine learning algorithms can be applied to the collected data to provide predictions, identify patterns, and offer recommendations to optimize agricultural practices. Examples include predictive crop yield modeling, disease detection, and pest management strategies.6.Mobile and Web ApplicationsMobile and web applications allow farmers to access real-time data, receive alerts, and control their agricultural systems remotely. These applications can provide visualizations, reports, and insights based on the analyzed data.7.Decision Support SystemThe decision support system utilizes the insights and recommendations generated by the analytics and machine learning layers to help farmers to make informed decisions about irrigation schedules, fertilizer application, crop rotation, and more.8.Security and PrivacyAs IoT systems deal with sensitive data, security measures must be in place to protect against unauthorized access and data breaches. Encryption, authentication, and access control mechanisms are implemented to ensure data privacy and integrity.

This extensive review shows that by implementing an IoT layer architecture for agriculture, farmers can improve their decision making, optimize resource utilization, and ultimately increase productivity. Moreover, the IoT- and DL-based techniques are more environmentally sustainable. For example, IoT-enabled CNN systems help in water resource management, pre-estimating fertilizer requirements, and early disease detection [44]. The works presented in [75] highlight the importance of collecting imagery datasets. They employed an appropriate DL model on the collected dataset for the detection and classification of plant diseases. They also focused on integrating DL models with IoT systems comprising sensors, a drone, a camera, etc. They claimed that these integrated systems reduce the human effort required. These systems can gather real-time information from farms and are important in the quick processing of the collected datasets to predict plant diseases. Similarly, Ref. [76] utilized the potential of IoT with DL. The authors implemented DHT11 sensors to measure temperature and humidity, to spot diseases at an early stage. These sensors were connected to a Raspberry Pi to transmit the captured information to the cloud server. This shows that IoT-enabled CNNs help in detecting climate changes causing sickness in crops. The importance of IoT in agricultural disease detection lies in its ability to provide real-time data, enable precision farming practices, and support informed decision making. By leveraging IoT technology, farmers can enhance their disease management strategies, improve crop yields, and contribute to sustainable and efficient agricultural practices. IoT technology allows farmers to remotely monitor their fields and livestock through mobile applications and web platforms. This capability is especially valuable for large-scale farms or when farmers are not physically present on the farm, enabling them to stay updated about their farm’s condition and respond quickly to potential disease outbreaks.

The authors in [77] proposed and implemented a framework integrating IoT and DL for pearl millet disease detection. In this research, the authors anchored the hardware components, such as sensors, drone cameras, and digital cameras, in the pearl millet farmland at ICAR, Mysore, India, to collect data automatically. The ‘Custom-Net’ model was designed as a part of this research and deployed on the cloud server. This DL model processes the data collected by the data collector and provides real-time prediction for blast and rust diseases.

## 4. Discussion

In this section, we elaborate on the analysis of the IoT and DL models applied for plant disease detection, visualization, and classification. We observe that factors such as the crop age, climate, and location affect the quality of the dataset and hence the accuracy of detection, visualization, and classification.

Based on the review of related works, we observe that IoT systems are important for the collection of real-time imagery as well as parametric datasets from fields. However, merely using IoT systems is not sufficient to obtain intelligent predictions about the nutrient, fertilizer, and pesticide requirements and disease detection and classification. Therefore, the datasets collected from IoT systems must be fed to DL models for analysis and decision making and to detect climate changes.

However, these integrated systems require improvements in terms of the linking of drones with GPS field sensors. GPS requires a continuous power supply, which increases the cost for farmers.

Moreover, it is observed that disruptions in the connectivity of sensors and cloud and mobile applications decrease the reliability of the integrated systems. Data storage and transmission via the cloud raise privacy and security issues. Such information can be disseminated as an alert or notification to farmers on their mobile phones.

Further, deficiencies in nutrients and diseases may leave similar visual symptoms on a crop plant. Thus, there is a need to design IoT systems that can capture even minor differences in texture, color, etc., in a cropped image, and the DL model should be optimized for the precise recognition of diseases and nutrient deficiencies. The symptoms are highly dependent on the season of growing, atmospheric conditions, and fertilization strategies. Thus, there is a need to design intelligent systems that can accurately predict diseases even for crops grown in variable environments.

Based on the review of DL models, we conclude that AlexNet minimizes the problem of vanishing gradients by using the ReLu activation function. Therefore, it can be applied for plant disease detection and classification. However, this model is inefficient in resource utilization, and it lacks parallel processing.

The model GoogleNet overcomes this drawback and employs parallel processing for the better utilization of resources. Moreover, it reports higher accuracy than AlexNet in the detection and classification of plant diseases. The DL models VGG-19, ResNet, and DenseNet report higher accuracy than GoogleNet on huge training datasets. However, these models require large storage space for the layers of the network and trainable parameters. Among all the models, DenseNet reports the highest accuracy in plant disease detection and classification, but it requires a long time for training.

We also observe that the models Inception, SqueezeNet, MobileNet, and modified or reduced MobileNet contain a smaller number of trainable parameters. These models require less storage space and shorter computational times. Therefore, these models may prove useful in providing mobile-based applications to assist farmers in predicting, detecting, and classifying plant diseases.

We also conclude that the hybrid models developed by employing VGG and Inception models or DenseNet and Inception models are storage-efficient and report high accuracy in disease detection, even in the case of complex backgrounds in images. A complex background refers to the presence of multiple objects, different colors, shadows, etc.

Further, we conclude that image segmentation and saliency mapping are useful for the visualization of plant diseases. However, hallenges such as the high log-loss, low accuracy for mobile applications, and imprecision in distinguishing diseases with similar symptoms need to be addressed.

The selection of the model and the fine-tuning of its hyperparameters according to the type of dataset is the key requirement in achieving the optimum performance of the model. Employing the ReLu or softmax activation function, adding a suitable number of dropout layers, and changing the number of layers in the neural network model according to the size and type of the dataset are important in achieving the optimum performance of the model. Applying pre-processing techniques such as rotation, histogram equalization, changes in the contrast of the image, etc., improve the robustness of the model and make it effective for datasets collected from different sources.

IoT holds the potential to revolutionize farming practices, increase productivity, enhance sustainability, and contribute to global food security. However, the successful implementation of IoT in agriculture requires that data privacy and security concerns be addressed. Moreover, we must consider the digital divide in rural areas and promote farmer education and awareness about the benefits of IoT adoption. Nonetheless, with ongoing technological advancements and industry collaborations, IoT is expected to play a crucial role in shaping the future of agriculture.

## 5. Conclusions

In this manuscript, we have successfully completed a systematic review of the literature to showcase the role of integrating cutting-edge technologies such as IoT and DL in automating agriculture processes. Based on the review, we conclude that the DL models VGG-19, ResNet, DenseNet, AlexNet, and GoogleNet report higher accuracy but are storage-inefficient. Thus, they cannot be integrated with mobile applications. Other DL models, such as Inception, SqueezeNet, MobileNet, hybrid models developed by employing VGG and Inception models, or DenseNet, require less storage without compromising accuracy. Therefore, these can be integrated with mobile applications for disease detection and classification. The hybrid models are also efficient in disease recognition from images with complex backgrounds. Thus, they would be useful in real-life implementation. Moreover, integrating IoT and DL models may prove useful in developing tools to assist farmers to improve the productivity and quality of crop products. Thus, it may reduce the costs of agriculture and revolutionize the area of plant disease detection, visualization, and classification. However, while DL- and IoT-based systems are available, there is huge scope to design power-efficient IoT devices for agriculture. Moreover, a great deal of research is required to improve the privacy, security, and non-interrupted communication of data between DL and IoT systems.

## Figures and Tables

**Figure 1 sensors-23-07877-f001:**
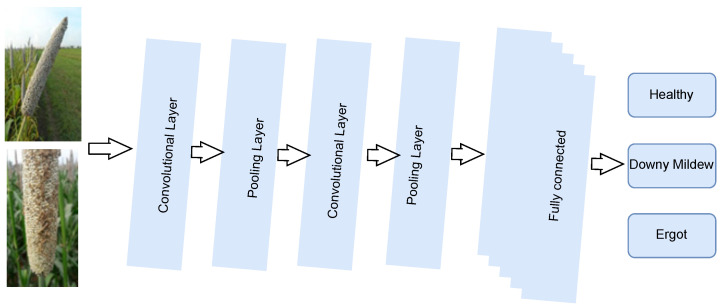
Architecture of IoT-enabled convolutional neural network.

**Figure 2 sensors-23-07877-f002:**
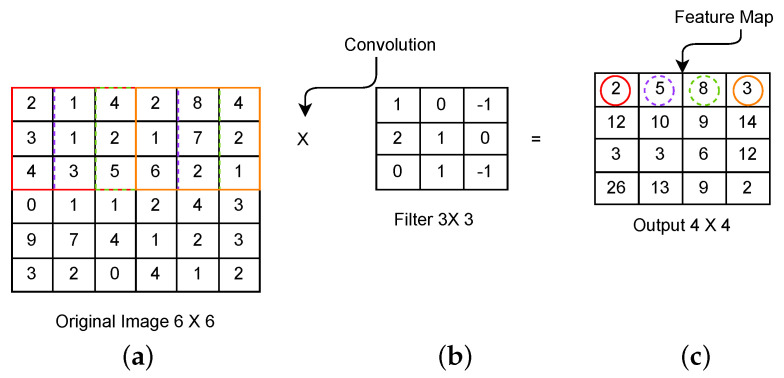
Convolutional operation process: (**a**) input matrix, (**b**) kernel, (**c**) feature map.

**Figure 3 sensors-23-07877-f003:**
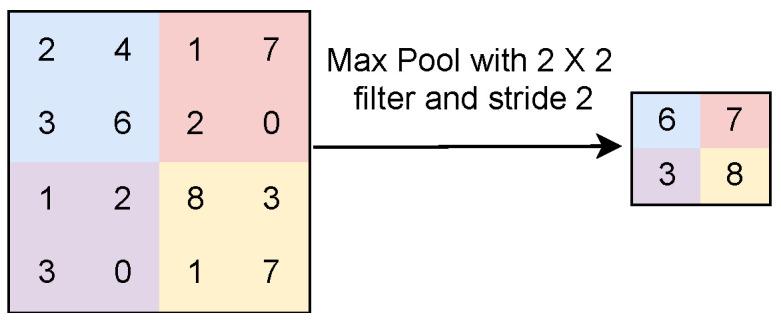
Max pooling operation.

**Figure 4 sensors-23-07877-f004:**
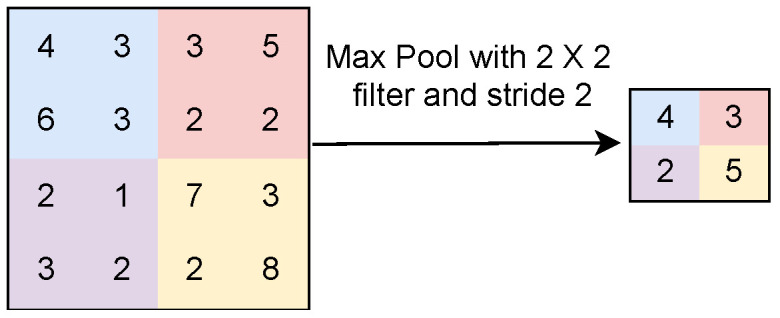
Average pooling operation.

**Figure 5 sensors-23-07877-f005:**
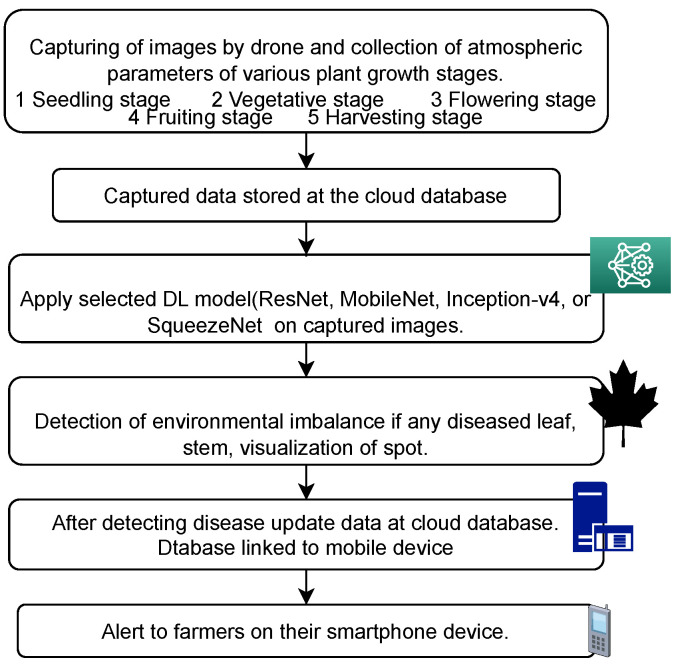
Flow of the proposed system.

**Figure 6 sensors-23-07877-f006:**
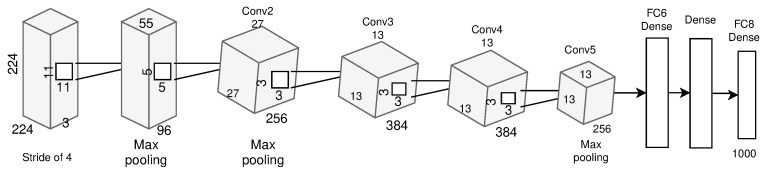
AlexNet architecture.

**Figure 7 sensors-23-07877-f007:**
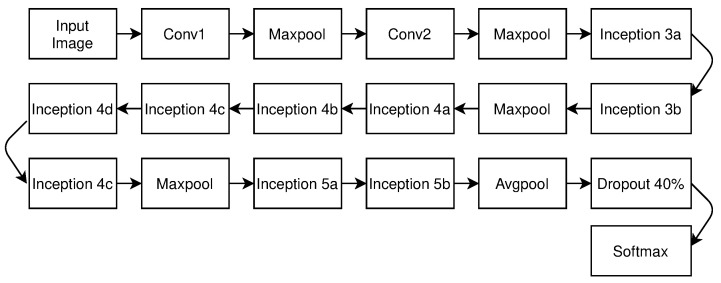
GoogleNet architecture.

**Figure 8 sensors-23-07877-f008:**
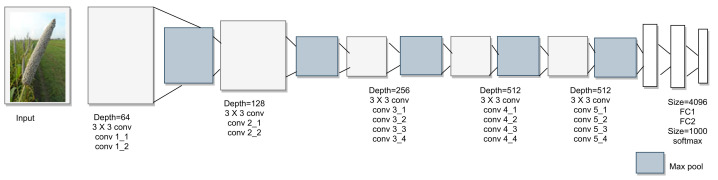
VGGNet architecture.

**Figure 9 sensors-23-07877-f009:**
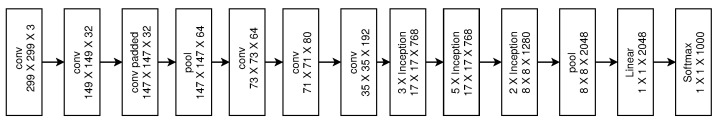
Inception-v3 architecture.

**Figure 10 sensors-23-07877-f010:**
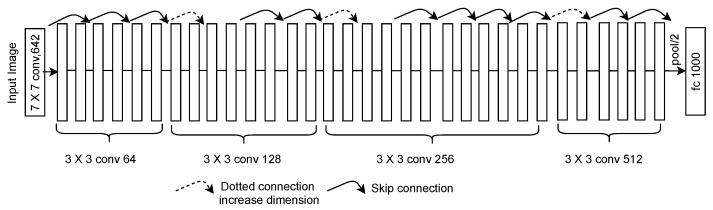
ResNet architecture.

**Figure 11 sensors-23-07877-f011:**
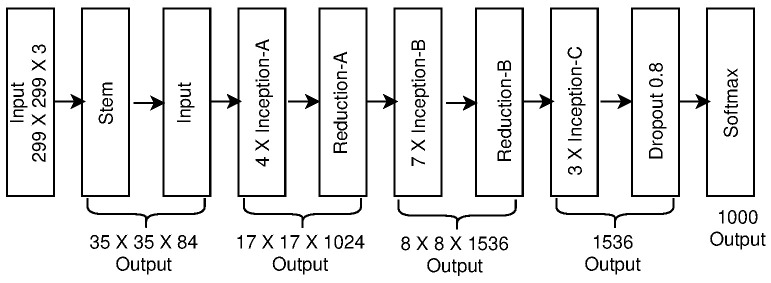
Inception-v4 and Inception-ResNet architecture.

**Figure 12 sensors-23-07877-f012:**
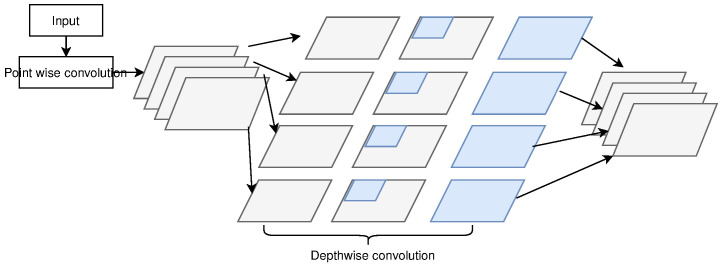
Xception architecture.

**Figure 13 sensors-23-07877-f013:**
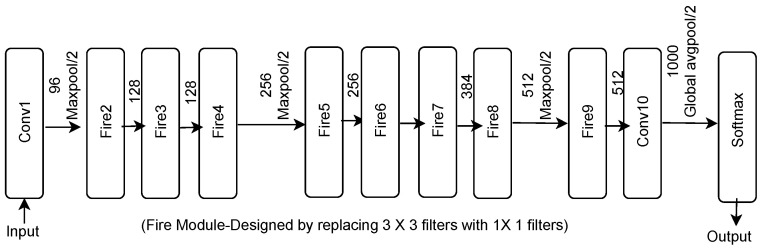
SqueezeNet architecture.

**Figure 14 sensors-23-07877-f014:**
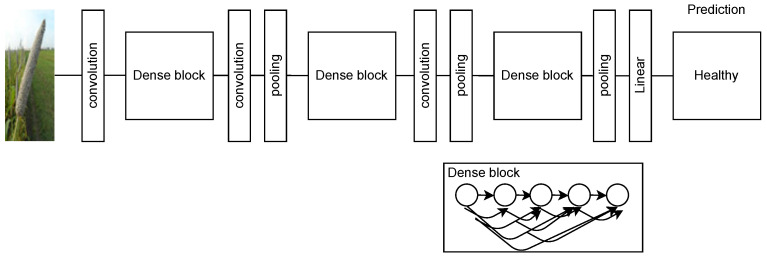
DenseNet architecture.

**Figure 15 sensors-23-07877-f015:**
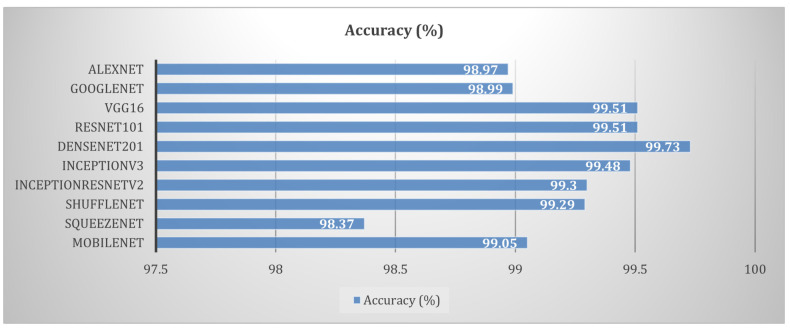
Performance comparison of deep learning models.

**Figure 16 sensors-23-07877-f016:**
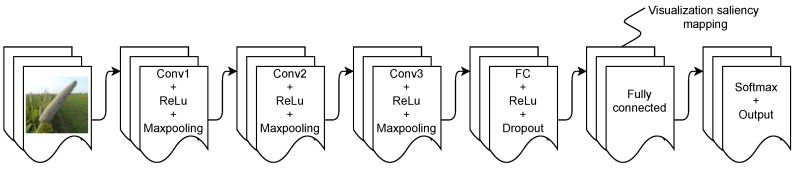
Modified architecture.

**Figure 17 sensors-23-07877-f017:**
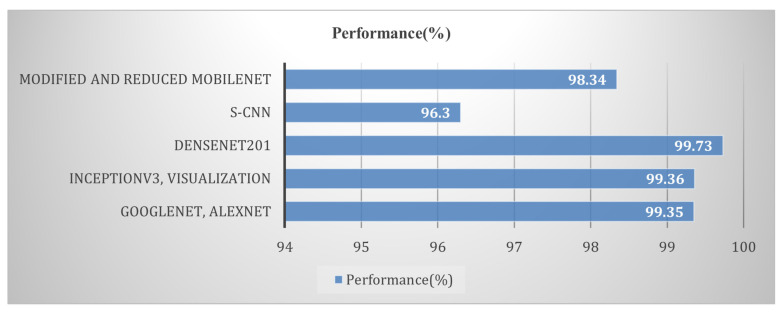
Performance comparison of deep learning models applied for plant disease detection and classification.

**Figure 18 sensors-23-07877-f018:**
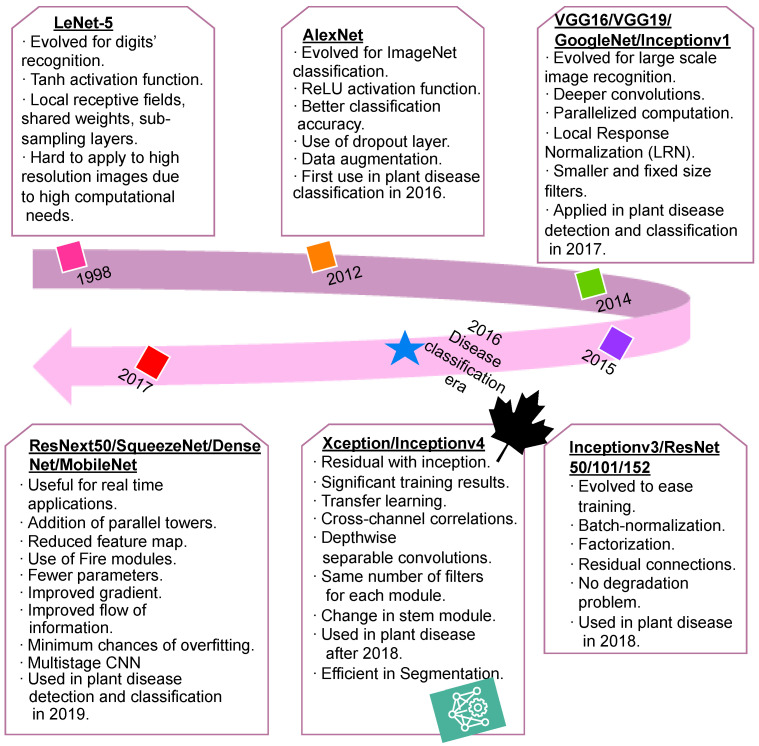
Deep learning era in plant disease research.

**Table 1 sensors-23-07877-t001:** Advantages and disadvantages of various DL models.

DL Model	Advantage	Disadvantage
AlexNet	Fast training because all perceptrons are not active together.	Shallow model that needs more time to achieve high detection accuracy.
GoogleNet/ Inception	Efficient in utilization of computing resources and showed low error rate of 6.67% in the ILSVRC-2014 competition. The error rate is 8.63% lower than that of AlexNet.	More prone to overfitting.
VGGNet-16 and VGGNet-19	High accuracy and fast training.	Prone to vanishing gradient problem.
Inception-v3	Requires smaller storage space. Achieves 2.5 times higher accuracy than GoogleNet.	Computationally expensive.
ResNet	Eight times deeper than VGG-19. Solves vanishing gradient problem. Error rate is 0.1% lower than that of Inception-v3.	Requires a long training time. Thus, it is difficult to apply for real-time problems.
Inception-v4 and Inception-ResNet	Hybrid of residual and inception networks.	Higher space and time complexity than ResNet-50, Inception-v3, and AlexNet models.
Xception	Inception modules replaced with depth-wise separable convolutions.	Memory- and time-intensive training.
SqueezeNet	Low computational complexity and high accuracy. Using FIRE modules achieves 50× smaller size than AlexNet.	Lower accuracy than larger and more complex models.
DenseNet	Collective information of the previous layer is transferred to the next layer. Thus, it creates direct connections among the intermediate layers and reduces the thickness of the layers.	Complex network models, excessive parameters, computationally and storage resource-intensive. Prone to overfitting.

**Table 2 sensors-23-07877-t002:** Comparison of ML and DL applied for disease detection and classification.

Crop(s) Selected	Disease Detected	Model Applied	Highest Accuracy
Wheat plant [56].	Powdery mildew, smut, black chaff, stripe rust, leaf blotch, leaf rust, healthy wheat.	VGG-FCN-VD16, VGG-FCN-S, deep learning, and multiple-instance learning.	VGG-FCN-VD16: 97.95%.
Rice [57].	Rice blast (RB), rice false smut (RFS), rice brown spot (RBS), rice bakanae disease (RBD), rice sheath blight (RSHB).	Gradient descent algorithm, the softmax learning algorithm, CNN in comparison with standard BP algorithm, SVM, particle swarm optimization.	Accuracy: 95.48% using CNN.
Mango [58].	Anthracnose.	Multi-layer convolutional neural network (MCNN), histogram equalization.	97.13% using MCNN.
Cucumber and apple PlantVillage [32].	38 classes of diseases.	AlexNet, GoogleNet Inception-v3, saliency maps.	99.67% using Inception-v3.
Cucumber [4].	Anthracnose, downy mildew, powdery mildew, and target leaf spots.	DCNN, random forest, SVM, AlexNet.	93.4% using the DCNN.
2108 images of citrus leaf [59]	Anthracnose, black spot, canker, scab, greening, and melanose.	Top-hat filter and Gaussian function, multicast support vector machine (M-SVM), contrast stretching method.	M-SVM gave accuracy of 97% on the citrus dataset and 89% on the combined dataset.
58 different species of plants [23].	Not specified.	VGGNet, Overfeat, AlexNet, AlexNetOWTBn, GoogleNet.	99.53% using VGGNet.
Maize [60].	Curvularia leaf spot, dwarf mosaic, gray leaf spot, northern leaf blight, brown spot, round spot, rust, and southern leaf blight.	GoogleNet, stochastic gradient descent (SGD) algorithm.	GoogleNet: 98.9%.
Apple plants [61].	Alternaria leaf spot, brown spot, mosaic, grey spot, and rust.	VGG-INCEP model, VGGNet-16, Inception-v3, Rainbow Single-Shot Detector, multi-box detector.	98.80% using the detection accuracy of Rainbow SSD multi-box detector.
Tea [62].	Tea red scab, teared leaf spot, and tea leaf blight.	Low-shot learning method, SVM, VGG-16, Conditional Deep Convolutional Generative Adversarial Network (C-DCGAN).	90% using VGG-16 and C-DCGAN.
Soybean leaves [63].	Downy mildew, spider mite.	ResNet, AlexNet, GoogleNet.	94.63% using ResNet.
Images of soybean stems [49].	Charcoal rot disease.	Saliency map, ResNet.	95.73% using ResNet.
Coffee, common bean, cassava, corn, citrus, wheat, sugarcane, passionfruit, soybean, grapevine, cashew nut, cotton, coconut tree, kale [64].	Multiple small or large spots, single large spot on the leaf, powdery spots.	GoogleNet, transfer learning.	94% using GoogleNet.

## Data Availability

This is a review article. Data sources used in the literature are mentioned in the text. No other data are associated with this article.

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
