# Peer review of "Role of Internet of Things and Deep Learning Techniques in Plant Disease Detection and Classification: A Focused Review"

_sensors, 2023, doi:10.3390/s23187877_

Round 1
Reviewer 1 Report
1. The serial numbers used in the introduction section should be distinguished from the serial numbers in the chapters in the text, namely lines 56, 72, and 82.
2. According to the sentence meaning, line 371 should be followed by line 370.
3. Table 1 should not be inserted in the middle of the paragraph.
4. Improve image quality, as shown in Figures 1 and 9.
5. It is necessary to briefly describe the content of each chapter of the article in the introduction section.
6. It is recommended to list the advantages and disadvantages of the deep learning model mentioned in Chapter 2 in a table, which can provide readers with a more intuitive understanding.
7. Figure 13 shows the experimental results of which crop in the plantvillage dataset?
8. The conclusion section of Chapter 5 is too lengthy, and a discussion section can be added. In addition, some English sentences have similar structures, and such problems should be avoided. Different sentence structures or words should be used as much as possible for expression.
Author Response
Role of Internet of Things and Deep Learning Techniques in Plant Disease Detection and Classification: A Focused Review
Response Sheet
We would like to thank Editorial and Reviewers’ team for sparing their valuable time to thoroughly review the manuscript and giving suggestions and comments to improve the manuscript. We carefully addressed all the comments and highlighted the major changes in yellow colour in the manuscript. The comment wise responses are discussed below. Kindly consider.
Reviewer #1
- Comment: The serial numbers used in the introduction section should be distinguished from the serial numbers in the chapters in the text, namely lines 56, 72, and 82.
Response: The serial numbers have been changed in the manuscript.
- Comment: According to the sentence meaning, line 371 should be followed by line 370.
Response: The content is ordered according to meaning in the manuscript.
- Comment: Table 1 should not be inserted in the middle of the paragraph.
Response: Correction done. Now, the table is not in the middle of the paragraph.
- Comment: Improve image quality, as shown in Figures 1 and 9.
Response: The quality has been improved.
- Comment: It is necessary to briefly describe the content of each chapter of the article in the introduction section.
Response: The content of each section is defined at the end of the introduction.
- Comment: It is recommended to list the advantages and disadvantages of the deep learning model mentioned in Chapter 2 in a table, which can provide readers with a more intuitive understanding.
Response: Table is added in section 2.
- Comment: Figure 13 shows the experimental results of which crop in the plant village dataset.
Response: The experiments were performed on the tomato crop dataset available in the PlantVillage dataset. The same is added to the manuscript.
- Comment: The conclusion section of Chapter 5 is too lengthy, and a discussion section can be added. In addition, some English sentences have similar structures, and such problems should be avoided. Different sentence structures or words should be used as much as possible for expression.
Response: As per the comment, the discussion section is added, and the conclusion section is reduced. Also, the English language-related editing is done throughout the manuscript.

Reviewer 2 Report
Overall, the manuscript provides a comprehensive review of the use of Internet of Things (IoT) and Deep Learning (DL) models for plant disease detection, visualization, and classification. The authors present their observations and conclusions based on the review of related works in the field.
Strengths:
1. Comprehensive Review: The manuscript covers various aspects of IoT and DL models applied to plant disease detection, including the impact of factors like crop age, climate, and location on dataset quality and accuracy.
2. Integration of IoT and DL: The authors emphasize the importance of integrating IoT systems with DL models, noting that this combination can be a game-changer in the field of plant disease detection and classification.
3. Identification of Challenges: The manuscript identifies several challenges such as disrupted connectivity, privacy, and security issues related to data storage and transmission via the cloud. It also highlights the need for precise recognition of diseases and nutrient deficiencies, especially in variable environments.
4. Evaluation of DL Models: The review provides insights into various DL models like AlexNet, GoogleNet, VGG-19, ResNet, DenseNet, Inception, SqueezeNet, MobileNet, etc., and their suitability for plant disease detection and classification.
5. Practical Recommendations: The manuscript offers practical suggestions, such as employing hybrid models, fine-tuning hyperparameters, and applying pre-processing techniques, to enhance the performance and robustness of DL models for different types of datasets.
Areas for Improvement:
1. Update on Current Trends: Since this review may have been conducted before the current date (July 2023), it is essential to ensure that the information and conclusions are up-to-date and relevant to the latest advancements in the field of IoT and DL for plant disease detection.
2. Practical Examples: Including practical examples or case studies of successful implementations of IoT and DL models in agriculture can further strengthen the manuscript's claims and provide real-world context.
3. Future Directions: The manuscript could include a section discussing future research directions and potential areas for improvement in the field of plant disease detection using IoT and DL.
In summary, the manuscript provides valuable insights into the use of IoT and DL models for plant disease detection, visualization, and classification. By addressing the areas for improvement, the authors can enhance the manuscript's clarity and impact, making it a valuable resource for researchers and practitioners in the agricultural domain.
Author Response
Role of Internet of Things and Deep Learning Techniques in Plant Disease Detection and Classification: A Focused Review
Response Sheet
We would like to thank Editorial and Reviewers’ team for sparing their valuable time to thoroughly review the manuscript and giving suggestions and comments to improve the manuscript. We carefully addressed all the comments and highlighted the major changes in yellow colour in the manuscript. The comment wise responses are discussed below. Kindly consider.
Reviewer #2
- Comment: Add current references with details in state of the art.
Response: Following references are added to show the current state-of-the-art.
- Shafi, U., Mumtaz, R., Shafaq, Z. et al. Wheat rust disease detection techniques: a technical perspective. J Plant Dis Prot 129, 489–504 (2022). https://doi.org/10.1007/s41348-022-00575-x.
- Saranya, C. Deisy, S. Sridevi, Kalaiarasi Sonai Muthu Anbananthen, A comparative study of deep learning and Internet of Things for precision agriculture, Engineering Applications of Artificial Intelligence.
- Gangwar, A.; Rani, G.; et al. Detecting Tomato Crop Diseases with AI: Leaf Segmentation and Analysis. In Proceedings of the 713 2023 7th International Conference on Trends in Electronics and Informatics (ICOEI). IEEE, 2023, pp. 902–907.
- Comment: One practical case study where combination of IoT and DL is implemented or can be implemented.
Response: Case study added at the end of section 4.
- Comment: Future scope to be added
Response: The future scope is added at the end of the conclusion section.

Reviewer 3 Report
The submitted manuscript is average and may be published after major revisions. The subject of the research is in line with the scope of the journal "Sensors".
Substantive remarks:
The content of the manuscript focuses on the issues of Convolutional Neural Network and Deep Learning. The Internet of Things issue is only mentioned in a few places, mainly in the "Introduction" chapter. The conclusions “we present an extensive review of the IoT, and DL models” (line 527) and “Based on the review of related works, we observed that IoT systems are important” (line 531) are not supported by the text of the manuscript.
The number of tables and figures to compare the different models is small. Table 1 is interesting, but many of the algorithms listed in the table are not explained in the manuscript.
In Chapter 2, “State-of-the-art deep learning models,” the authors compare 9 deep learning models. The 5 models are illustrated with schematics of completely different styles. The other models don't have their own schematics. It is worth presenting schematics of all models using the same style. This will show the differences in the models.
Figure 11 is based entirely on the data published in [48]. The data source is not specified in the description of the figure.
The figures contain numerous errors in the description, e.g. Fig. 4 - description above the arrow, Fig. 6 - wrong direction of the arrows, Fig. 14 - dates on the ribbon are not related to the content of the drawing.
Author Response
Role of Internet of Things and Deep Learning Techniques in Plant Disease Detection and Classification: A Focused Review
Response Sheet
We would like to thank Editorial and Reviewers’ team for sparing their valuable time to thoroughly review the manuscript and giving suggestions and comments to improve the manuscript. We carefully addressed all the comments and highlighted the major changes in yellow colour in the manuscript. The comment wise responses are discussed below. Kindly consider.
- Comment: The content of the manuscript focuses on the issues of Convolutional Neural Network and Deep Learning. The Internet of Things issue is only mentioned in a few places, mainly in the "Introduction" chapter. The conclusions “we present an extensive review of the IoT, and DL models” (line 527) and “Based on the review of related works, we observed that IoT systems are important” (line 531) are not supported by the text of the manuscript.
The number of tables and figures to compare the different models is small. Table 1 is interesting, but many of the algorithms listed in the table are not explained in the manuscript.
Response: For more justification one more table added in section 2 to show comparison of various DL models. Also, figures of remaining DL models are added to the manuscript.
- Comment: In Chapter 2, “State-of-the-art deep learning models,” the authors compare 9 deep learning models. The 5 models are illustrated with schematics of completely different styles. The other models don't have their own schematics. It is worth presenting schematics of all models using the same style. This will show the differences in the models.
Response: Now, schematics and comparison of remaining models is added to the manuscript.
- Comment: Figure 11 is based entirely on the data published in [48]. The data source is not specified in the description of the figure.
Response: The data source is mentioned before Figure 11 in the manuscript and highlighted.
- Comment: The figures contain numerous errors in the description, e.g. Fig. 4 - description above the arrow, Fig. 6 - wrong direction of the arrows, Fig. 14 - dates on the ribbon are not related to the content of the drawing.
Response: Figures 4 and 6 are updated, Dates in Figure 14 are as per the history of the DL models in the disease detection era.
